

# Emergence of a localized total electron content enhancement during the G4 geomagnetic storm of September 8, 2017

Carlos Sotomayor-Beltran[1]

[1]Image Processing Research Laboratory (INTI-Lab), Universidad de Ciencias y Humanidades, Lima 39, Peru

**Correspondence:** Carlos Sotomayor-Beltran (csotomayor@uch.edu.pe)

**Abstract.**

In this work, the first results of the analysis on total electron content (TEC) data before, during and after the geomagnetic storm of September 8, 2017 are reported. A common response to geomagnetic storms due to the southern vertical interplanetary magnetic field ($B_z$) is the enhancement of the electron density in the ionosphere. Vertical TEC (VTEC) from the Center

for Orbit determination in Europe (CODE) along with a statistical method were used to identify positive and/or negative ionospheric storms in response to the geomagnetic storm of September 8, 2017. When analysing the response to the G4 storm of September 8, 2017 it was indeed possible to observed an enhancement of the equatorial ionization anomaly (EIA); however what it is was unexpected, was the identification of a local TEC enhancement (LTE) to the south of the EIA ($\sim 40°$ S, right over New Zealand and extending towards the south-eastern coast of Australia and also eastward towards the Pacific). This was

a very transitory LTE that lasted approximately 2 hours, starting at $\sim$02:00 UT on September 8 where its maximum VTEC increase was of 241,2%. Using the same statistical method we looked for LTEs in a similar category geomagnetic storm, the G4 storm of St. Patrick's day of 2015; however for this storm, no LTEs were identified. As also indicated in a past recent study for the August15, 2015 geomagnetic storm, an association between the LTE and the excursion of $B_z$ observed during the September 8, 2017 storm is observed. Nevertheless, it is more likely that a direct impact of the super-fountain effect along

with another ionospheric physical mechanism may be playing an important role in the production of this LTE.

## 1 Introduction

Anomalies in the ionosphere can be product of different natural phenomena (Afraimovich et al., 2013). For instance, earthquakes can produce positive or negative ionospheric anomalies (e.g., Zakharenkova et al., 2008; Yao et al., 2012; Guo et al., 2015; Li et al., 2015). Although, such variations are expected to be localized within the earthquake's preparation region (Dobro-

volsky et al., 1979). On the other hand, major changes in the ionosphere are caused by geomagnetic storms (e.g., Buonsanto, 1999; Danilov, 2013). The response of the Earth's ionosphere to the geomagnetic storms are known as ionospheric storms. These ionospheric storms can disrupt technologies relying on transmission of radio frequencies (e.g., Buonsanto, 1999; Borries et al., 2015), and thus they can have an impact in the modern society in general.

In order to understand better ionospheric variability in time and space produced by geomagentic storms, Global Navigation

Satellite System (GNSS) receivers, due to its global coverage, are used as one of the main tools for ionospheric studies.





According to several studies (e.g., Huang et al., 2005; Mannucci et al., 2005; Astafyeva, 2009), one common response to a geomagnetic storm due to the excursion of the southward interplanetary magentic field is the significant increment in the equatorial and mid-latitude total electron content (TEC), which manifests as an enhancement of the equatorial ionization anomaly (EIA; Appleton, 1946; McDonald et al., 2011). Such increase of TEC in the EIA is possible to visualize in global

ionospheric maps (GIMs). Besides changes in the EIA, it was recently observed by Edemskiy et al. (2018) that localized TEC enhancemenets (LTEs) can also emerge as a response to a geomagentic storm.

In this paper vertical TEC maps, also known as global ionospheric maps (GIMs), due to its reliability on ionospheric infor- mation (Hernández-Pajares et al., 2009), were used to analyze the response to the geomagnetic storm of September 8, 2017. Section 2 introduces the ionospheric data and and the technique for the corresponding analysis. In Sect. 3 the results and the

discussion are presented. Section 4 presents the final remarks or conlcusions.

## 2   Data and methods

VTEC maps were downloaded via ftp[1] from the Center for Orbit Determination in Europe (CODE) between August 21, 2017 and September 20, 2017. VTEC maps, which have a resolution of 2.5° x 5° (latitude and longitude, respectively), come in daily IONnosphere Map EXchange files (Schaer et al., 1998) and they are produced every hour. Due to the format of the IONEX

files, which consists of headers and the actual VTEC data, a code entirely written in Python was implemented for this work. Using the NumPy[2] library, which handles relatively easily N-dimensional arrays, the VTEC data was stored in a 3D cube for further analysis. The $x$, $y$ and $z$ axes in the 3D cube are longitude, latitude and number of maps, respectively.

In order to indentify ionospheric anomalies we apply a running window of 8 days to every cell in the 3D VTEC cube (e.g., Liu et al., 2004; Zhu et al., 2010; Zou and Zhao, 2010; Li et al., 2015). Within this window the median ($\bar{X}$) and the interquartile

range (IQR) are calculated to define the upper and lower bounds. However, assuming that the VTEC data follows a normal distribution within the window, the upper and lower bounds can be defined as:

$$UB = \mu + 2\sigma, \tag{1}$$

$$LB = \mu - 2\sigma, \tag{2}$$

were $\mu$ and $\sigma$ are the mean and standard deviation, respectively. If a VTEC value for a certain day at a particular time falls

above the $UB$, then a positive ionospheric anomaly is detected with a confidence level of 95%. The difference between the VTEC and $UB$ or $LB$ is defined as differential VTEC ($\Delta$VTEC). On the other hand, if the VTEC falls bellow the $LB$, then a negative anomaly is detected. In this way, a cube of $\Delta$VTEC is created, with a total of 744 maps. If $UB >$ VTEC $> LB$, then $\Delta$VTEC $= 0$

---

[1]ftp://ftp.aiub.unibe.ch/CODE/
[2]http://www.numpy.org/



Some important geomagnetic parameters are also needed to be taken into account for the analysis. The Dst index (Sugiura, 1964) provides information about the strength of the ring current around the Earth. According to Loewe and Prölss (1997) a magnetic storm can be considered as weak when -50 nT $<$ Dst $\leq$ -30 nT. A moderate and strong storm occurrs when -100 nT $<$ Dst $\leq$ -50 nT and -200 nT $<$ Dst $\leq$ -100 nT, respectively. Finally, a severe storm happens when Dst $\leq$ -200 nT. For

this study Dst data for the month of September 2017 was downloaded from World Data Center for Geomagnetism in Kyoto[3]. Another very important index which measures the fluctuations caused in the Earth's magnetic field by a geomagnetic storm is the Kp index. According to Gosling et al. (1991) when Kp $\geq$ 8- and Kp $\geq$ 6- for at least three 3-h intervals, the storm can be considered a major one. A large storm occurs when 7- $\leq$ Kp $\leq$ 7 and Kp $\geq$ 6 for at least three 3-h intervals. For other cases when Kp $\geq$ 6- for at least three 3-h intervals the storm can be considered of medium strength. Finally, a small storm

happens when -5 $\leq$ Kp $\leq$ 5. Kp data for September 2017 was retrieved from the German Research Centre for Geosciences (GFZ[4]). The vertical interplanetary magnetic field ($B_z$; Tsurutani et al., 1988) also is a good indicator of a geomagnetic storm. When there is a strong southward $B_z$ for more than 3 hours a geomagnetic storm is in development (Gonzalez et al., 1994; Liu and Li, 2002). Hourly averages from $B_z$ where dowloaded from the OMNI datasabe[5]. In Fig. 1 the Dst and Kp indices and also the southward interplanetary $B_z$ (in geocentric solar magnetospheric coordinate system) can be observed for the month of

September 2017.

## 3 Results and discussion

Figure 1 shows that Kp = 8 during the last 3 hours (UT) of March 7 and the first three hours of March 8. According to the National Oceanic and Atmospheric Administration (NOAA) space weather service[6], this geomagnetic storm can be classified as a G4 severe storm (Kp = 8). Additionally, for March 8, 2017 between 00:00 and 04:00 UT the Dst index had values lower

than -100 nT (Fig. 1).

The origin of this geomagnetic storm lies in the coronal mass ejection (CME) that occurred on September 6, 2017 at $\sim$12:40 UT. This CME was observed with the Camera 2 of the Large Angle and Spectrometric Coronograph on board of the Solar and Heliospheric Observatory (SOHO[7]). Figure 1 also shows that on September 8 at $\sim$00:00 UT the vertical interplanetary magnetic field decreased significantly to a minimum of -24 nT. One hour before (September 7 at 23:00 UT), $B_z$ already

decreased considerably to -20.6 nT, time of the storm sudden commencement (Fig. 1). In addition, it can be noticed that almost simultaneously with the drastic change of $B_z$, the Dst index reached its peak at  01:00 UT on September 8, 2017. This relationship between $B_z$ and the Dst index hints to a physical response of the ring current in the magnetosphere to the interplanetary field $B_z$ (Patel and Desai, 1973; Gonzalez and Echer, 2005).

---

[3]http://wdc.kugi.kyoto-u.ac.jp/wdc/Sec3.html
[4]https://www.gfz-potsdam.de/en/kp-index/
[5]https://omniweb.gsfc.nasa.gov/form/dx1.html
[6]https://www.swpc.noaa.gov/noaa-scales-explanation
[7]https://sohowww.nascom.nasa.gov/



In the right column of Fig. 2, GIMs for September 7, 8 and 9, 2017 at 02:00 UT are presented. It is clearly seen in the GIM of September 8 at 02:00 UT (just three hours after the storm sudden commencement) that the VTEC was enhanced in the EIA region with respect to the day before (September 7) and the day after (September 9) at the same hour. This increment of VTEC was already observed in previous studies about ionospheric responses to geomagnetic storms (e.g., Zhao et al., 2005; Pedatella et al., 2009; Astafyeva et al., 2015; Chakraborty et al., 2015). Moreover, a ionospheric localized anomaly (∼40° S), or as named by Edemskiy et al. (2018) a localized TEC enhancement (LTE), to the south of the southern conjugate geomagnetic region of the EIA was identified in the GIM map of September 8, 2017 at ∼02:00 UT. This LTE was very transitory, in the ΔVTEC maps it appeared at ∼02:00 UT on September 8 and at ∼04:00 UT it was already gone. In the left column of Fig. 2, ΔVTEC maps for September 7, 8 and 9, 2017 at 02:00 UT are also presented. It can be seen from these ΔVTEC maps that a day before and after that the LTE appeared, no anomalies were visible. However as already indicated, the day that the ionospheric storm occurred (September 8), the dramatic enhancement of the VTEC to the south of the EIA, manifested as a LTE, is observed. This unforseen positive ionospheric storm covers most of New Zealand and extends westward towards the south-eastern part of Australia and eastward towards the Pacific. The maximum peak of this LTE happened as well on September 8 at 02:00 UT with ΔVTEC = 6.47 TECU (where 1 TECU = $10^{16}$ electrons/m$^2$). To better visualize this LTE to the south of the EIA, the shape of the VTEC along the meridional line of 170°E is shown in Fig. 3 between September 7 and 9, 2017 at 02:00 UT. From the ΔVTEC maps, it can be confirmed that the EIA follows its normal variability one day after (September 9 at 02:00 UT) and before (September 7 at 02:00 UT) that the storm occurred (no anomalous VTEC enhancements are visible). However, on September 8 at 02:00 UT the EIA is significantly enhanced and hence this translates in a much sharper definition of the double-crest with a trough shape observed in Fig. 3. This shape is expected because when the LTE is above New Zeland, it is still day time, the local time is 14:00 (02:00 UT). In addition to the two crests from the EIA, a third one in the southern hemisphere is visible (Fig. 3). This third crest is simply the LTE observed in the ΔVTEC and GIM maps for September 8 at 02:00 UT. The peak increment for this day and this time in the southern crest of the EIA is of 172% and in the LTE of 241,2%. Edemskiy et al. (2018) has also reported for the August 15, 2015 G2 geomagnetic storm that the two LTEs they observed were located to the south of the EIA (between Africa and Antarctica).

In order to look for such LTEs in a similar geomagnetic storm category, the author turn to the G4 geomagnetic storm that happened during the St. Patrick's day of 2015, which has been thoroughly studied (Astafyeva et al., 2015; Cherniak et al., 2015; Nava et al., 2016; Yao et al., 2016; Jin et al., 2017; Zhang et al., 2018). This storm was also product of a CME an it was reported that the storm sudden commencement was at ∼04:45 UT on March 17, 2015 (Yao et al., 2016). The statistical method applied to the G4 storm of September 8, 2017 in this paper was also applied to GIMs during the St. Patrick's storm. IONEX files from CODE were downloaded and processed with the Python software written for this work for the range of days between February 27, 2015 and April 3, 2015. Part of the resultant ΔVTEC maps are shown in Fig. 4. Going through the ΔVTEC maps created for the aforementioned range of days, it was possible to observe a positive ionospheric storm starting on March 17, 2015 at ∼18:00 UT right over the southern Atlantic, right north off the Antarctic coast. This positive storm started to move westward and it reached its maximum strength on March 18, 2015 at ∼02:00 UT with a peak of ΔVTEC = 12.88 TECU (Fig. 4). In this case however, the enhancement of VTEC observed in the southern hemisphere is not a LTE, it is only the sourthern crest





of the EIA which underwent an increment of VTEC and shifted some degrees southward. On the other hand in the ΔVTEC maps of March 17, 2015 starting at ∼22:00 UT, negative ionospheric storms were also observed and they lasted until the end of the day of March 18, 2015. These both results agree well with the ones from previous studies, using different methods, for the St. Patrick's day 2015 storm (Astafyeva et al., 2015; Yao et al., 2016). It can also be finally noticed in Fig. 4 in the ΔVTEC

maps that at 02:00 UT the day before and the day after the maximum peak of the positive ionospheric storm, the increment or decrement of VTEC are minimal.

For the case of the St. Patrick storm of 2015, when the observed positive storm in the southern hemisphere and negative storm in the northern hemisphere are co-existing (what is also known as hemispheric asymmetry), it could be assumed that the mechanism at work producing this asymmetry was the storm-time thermospheric circulation (Fuller-Rowell et al., 1994;

Fang et al., 2012). However according to this theory, the positive ionsopheric storms are expected in the winter hemisphere and the negative ionospheric storms in the summer hemisphere; hence, Astafyeva et al. (2015) and Yao et al. (2016) ruled out this theory as a possibility for the origin of the detected ionospheric storms. They, nevertheless, indicated three more suitable candidates: the strength of the geomagnetic field, the $B_y$ component of the interplanetary magnetic field and composition changes in the thermosphere. On the other hand, for the moderate G2 storm of August 15, 2015 (Edemskiy et al., 2018) there

was not a clear mechanism put forward to account for the observed LTEs. Only a dependance of the emergence of these LTEs to the interplanetary $B_z$ was hinted at, but still as indicated by the authors of that study it was not their definite conclusion. For the LTE observed during the September 8, 2017 severe storm in this work, an excursion the interplanetary $B_z$, along with a consequent decrease of the Dst index, was also observed (Fig. 1). Thus, it can be suggested that there is as well an association between the interplanetary $B_z$ and the emergence of the LTE. As per to the overall enhancement of the EIA (Fig. 2 and 3)

and shifting of the crests in the direction of the poles observed in Fig. 3, it is suggested by many studies (e.g., Tsurutani et al., 2004; Mannucci et al., 2005; Astafyeva, 2009; Astafyeva et al., 2014; Chakraborty et al., 2015) that the mechanism at work for this change of shape of the EIA is the ionospheric super-fountain effect. How this effect is connected or contributes to the appeareance of the LTE observed on September 8, 2017 at ∼02:00 UT is still not clear.

## 4 Conclusions

Ionospheric response to the G4 severe geomagnetic storm of September 8, 2017 was analysed by using VTEC maps from CODE along with a statistical method to identify ionospheric anomalies. By producing differential VTEC maps it was possible to identify not only an enhancement of the EIA but also a localized TEC enhancement. The maximum intensity of this LTE was on September 8, 2017 at 02:00 UT and it was localized right over New Zeland and extending towards the south-eastern coast of Australia and eastward towards the Pacific. The LTE was quite transitory, it lasted only about two hours, on September

8 at 04:00 UT it faded away. By analyzing the latitudinal profiles, it could be determined that the increment in intensity for this LTE was of 241.2%.

Due to its category, the G4 storm from March 17, 2015 was also investigated in order to look for LTEs; however, there was no LTE detections. What is was discovered was a hemispheric asymmetry of ionospheric storms in the northern and southern




hemisphere. The origin of this asymmetry was explained in past studies by the strength of the geomagnetic field, the $B_y$ component of the interplanetary magnetic field and composition changes in the thermosphere.

One geomagnetic storm which presented the same traits (LTEs) as in the one of Spetmeber 8, 2017 was the G2 August 15, 2015 moderate storm. During this storm also LTEs were identified south of the geomagnetic conjugate region of the EIA. These

5  LTEs, was indicated, seem to be associated with the negative excursion of $B_z$. Because for the September 8, 2017 storm also such negative excursion was observed, it can be suggested then that the vertical interplanetary magnetic field component has an effect on the origin of the LTE. However, due to the fact that the EIA undergoes a dramatic enhancement, the contribution of the super-fountain effect in the generation of the LTE would have to be taken into account as well. To shed more light into how these LTEs are created, further observations of these events along with physical modeling of the effects of the $B_z$ on the

10  super-fountain effect and possibly other contributing ionospheric mechansims would be needed.

*Competing interests.* The author declares that he does not have conflict of interest.

*Acknowledgements.* The author is very thankful to CODE for making publicly available IONEX files. The author would like also to ac-knowledge the World Data Center for Geomagentism in Kyoto, the German Research Center for Geosciences and the OMNI database for providing data for the Dst index, the Kp index and the vertical interplanetary $B_z$, respectively.





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





**Figure 1.** The Dst and Kp indices and the southward interplanetary magnetic field ($B_z$) for the month of September 2017. The vertical red dashed line in all the plots points to the storm sudden commencement.

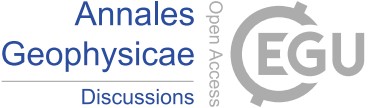



**Figure 2.** Left column: Differential VTEC maps for September 7, 8 and 9, 2017 at 02:00 UT. Right column: Global ionospheric maps for September 7, 8 and 9, 2017 at 02:00 UT.



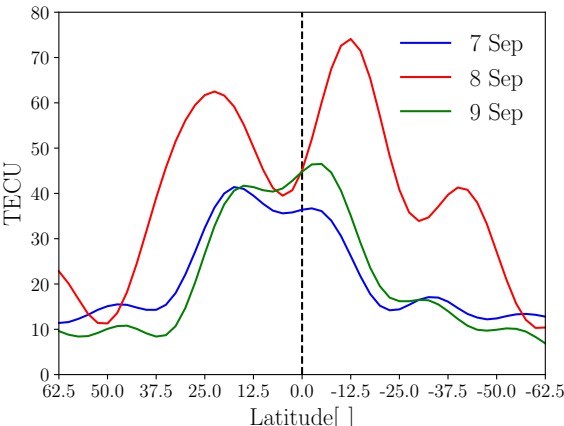

**Figure 3.** Structure of the VTEC for the 170°E meridian at 02:00 UT between September 7 and 9, 2017. A relevant range of latitudes is shown, 62.5°N–62.5°S. The vertical dashed black line indicates the Equator (latitude = 0°).





**Figure 4.** Left column: Differential VTEC maps for March 17, 18 and 19, 2015 at 02:00 UT. Right column: Global ionospheric maps for March 17, 18 and 19, 2015 at 02:00 UT.