# Peer review of "Emergence of a localized total electron content enhancement during the severe geomagnetic storm of September 8, 2017"

_Annales Geophysicae, 2018_

## Author Comment (AC1) · 13 Jul 2018

Due to some unknown reasons the x tick labels in Figure 1 and the North(N), South(S), West(W), and East(E) symbols in the maps of Figures 2 and 4 are missing. Thus, here I am uploading Figures 1, 2, and 4 to show the details that are missed.

[Figure]

[Figure]

**Fig. 1.** Figure 1 in the manuscript

[Figure]

**Fig. 2.** Figure 2 in the manuscript

[Figure]

**Fig. 3.** Figure 4 in the manuscript

---

## Referee Comment (RC1) · Anonymous Referee #1 · 15 Aug 2018

Reviewer's comments to the paper by Sotomayor-Beltran "Emergence of a localized total electron content…"

Total electron content (TEC) enhancements during ionospheric storms of 2017 and 2015 are analyzed in the paper. The author uses his own method of determining deviations in TEC during a storm from quiet conditions. In the majority of ionospheric storm studies, the deviations in foF2 or TEC are studied comparing observed values of the studied parameter with its values during the preceding quiet days, or with a median. The author presents a brief discussion of the method used in the paper (Section 2), however the description is not clear. As far as I understand, for each spatial sell of the data, the 8-day running window is used to calculate the median (X). However, the median is not mentioned later in the text. The formulae (1) and (2) for the upper and lower bounds (UB and LB, respectively) relate UB and LB to $\mu$ and $\sigma$ (UB = $\mu+\sigma$ and LB = $\mu-\sigma$), "… were $\mu$ and $\sigma$ are the mean and standard deviation, respectively". One could understand form this determination that $\mu$ is a mean deviation. However then formulae (1) and (2) became senseless, because UB and LB would have a dimension of errors, but not of absolute values of TEC. If the author means that $\mu$ is a mean value, then it is not clear how it has been obtained. Probably, X should stand in formulae (1) and (2) instead of $\mu$. Then at least, the formulae would be understandable.

The description of the results begins from an error. In the first paragraph of Section 3, Figure 1 is considered. In this paragraph, March 7 and March 8 are mentioned while considering this figure. However, it follows from the caption to Figure 1 that the figure contains data for September, 2017. Obviously, March 7 and March 8 in the text should be September 7 and September 8, respectively.

Figure 1 does not have dates at the abscissa (only numerals 2), so it is impossible to relate the behavior of geomagnetic and interplanetary indices to UT and dates and to compare this behavior with the TEC data shown in Fig. 2.

The main result is that the author has found during the September 2017 storm not only an enhancement of the equatorial ionospheric anomaly, but also a localized TEC enhancements (LTE). No such enhancements are detected during the storm of March 2015, which is of the same intensity as the September 2017 storm. However, the presence of LTE was detected during the August 15, 2015 storm of lower intensity.

Since ionospheric storms are very different from each other in the ionospheric behavior in time and space, I think that information on foF2 or TEC behavior during any storm is valuable and deserves publication, but only if it is presented in the proper form. The results obtained in this paper could be published, but the paper should be substantially revised to make the publication possible.

Besides the comments made above, I think that a figure similar to Figure 1 should be included for the March 2015 storm in order to make it possible to compare the data in Fig. 4 with the behavior of geomagnetic and interplanetary indices.

The language of the paper is poor and needs a serious improvement.

My recommendation is a major revision.

---

## Referee Comment (RC2) · Anonymous Referee #2 · 27 Aug 2018

Referee report on the paper "Emergence of a localized total electron content enhancement during the G4 geomagnetic storm of September 8, " by Carlos Sotomayor-Beltran

The paper is devoted to the study of the ionospheric storm, using total electron content data occurred on 7-9 September 2017. In particular the author put in evidence what he calls a localized total electron content enhancements, and increase of TEC respect a background, at Southern mid latitude hemisphere.

General Comments

Abstract A common response to geomagnetic storms due to the southern vertical interplanetary magnetic field (Bz) is the enhancement of the electron density in the iono-

sphere

This statement general is incorrect. Not all ionospheric storms start with a positive phase. The storm pattern depends on season, longitudinal sector, the intensity of a geomagnetic storm, LT of a magnetic storm commencement.

Looking at the figure 2 I see that the increase of TEC is more extended versus the mid-latitude in the Southern hemisphere but it is possible to see it also in the northern hemisphere. So for my point of view is not a localized enhancement .Looking at the paper by Lei et al. (2017 - 0.1029/2017JA02516) that analyze the same event with TEC and ionosonde data and in their figures it can be seen an increase of differential TEC for all latitudes in the first hours of 8 September, that corresponds to the storm main phase, in the northern hemisphere and in the southern also( but they arrive at 24 degree in latitude)that correspond to a positive ionospheric storm. In the following days a negative ionospheric storm occurred from higher to lower latitude. So it is a typical ionospheric storm with a positive and negative phase , the physical mechanism for the positive phase occurred at 02UT(daytime) that the author identify as a localized event seeing only the figure at 2UT could be due to an expanded convection electric field during geomagnetic storms in these cases frequently it is observed dTEC enhancements in the mid latitude dayside ionosphere but more investigations are necessary.

L 6 P1 What is "the G4 storm"? G4 should be explained

L8 P1 what it is was unexpected Grammar

L26 P1 Global Navigation Satellite System (GNSS) receivers, due to its global coverage, are used as one of the main tools for ionospheric studies. This is not so. The whole morphology of ionospheric storms has been obtained and understood using the world-wide ground-based ionosonde network observations. Only vertical ionospheric sounding gives directly electron concentration in the ionospheric layers. VTEC on one hand is obtained from slant TEC observations on the other hand it includes the plasmaspheric part which is not related to the underlying ionosphere. For this reason VTEC

may be only considered as a complementary source of information for such type of analysis.

Introduction should formulate the problem which is solved in the paper rather than general words related to ionospheric variations. The increase of daytime plasma uplift from the geomagnetic equator due to penetrating electric field was discussed repeatedly in the past, but the author did not mention all these publications. So, this is not clear what is a new ionospheric effect discussed in the paper in a comparison to what is already known.

L 18 P2 ...we apply a running window of 8 days... Why 8 day window? What is the idea for such choice? What to do with such background if these 8 previous days were disturbed?

L 17 P3 Figure 1 shows that Kp = 8 during the last 3 hours (UT) of March 7 and the first three hours of March 8. March has not been discussed yet in the paper.

A positive storm phase (the first phase) of a two-phase ionospheric storm is a normal reaction of the day-time mid-latitude ionosphere to a strong geomagnetic storm (started in the daytime sector). Some examples and mechanisms may be found in J. Atmos. Solar-Terr. Physics., 81-82, 59-75, 2012." Two types of positive disturbances in the daytime mid-latitude F2-layer: Morphology and formation mechanisms".

L 1 P 5 Looking at the figure 2 I see that the increase of TEC is more extended versus the mid-latitude in the Southern hemisphere but it is possible to see it also in the northern hemisphere. So for my point of view is not a localized enhancement .Looking at the paper by Lei et al. (2017 - 0.1029/2017JA02516) that analyze the same event with TEC and ionosonde data and in their figures it can be seen an increase of differential TEC for all latitudes in the first hours of 8 September, that corresponds to the storm main phase, in the northern hemisphere and in the southern also( but they arrive at 24 degree in latitude)that correspond to a positive ionospheric storm. In the following days a negative ionospheric storm occurred from higher to lower latitude. So it is a typical

ionospheric storm with a positive and negative phase , the physical mechanism for the positive phase occurred at 02UT(daytime) that the author identify as a localized event seeing only the figure at 2UT could be due to an expanded convection electric field during geomagnetic storms in these cases frequently it is observed dTEC enhancements in the mid latitude dayside ionosphere but more investigations are necessary.

In general the paper does not present either any new morphological effect or physical interpretation. I cannot recommend this paper publication.

---

## Author Response (AR1)

Dear Editor,

I would like to sincerely thank you for you valuable comments in an effort to improve my article.

In the revised version I have already addressed your concerns:

1) From my side, I would like to point out that the space weather event you analysed, was very interesting, but complicated. Four days-long period before the storm maximum was under the diminishing influence of a positive polarity coronal hole high speed stream (CH HSS), when solar wind speed ranged from 430 to 680 km/s with total field between 3-9 nT. According to the warning issued by NOAA, the geomagnetic field was already at the active levels on 5-6 September. Total field increased twice, for the first time it increased to 16 nT at 6 of September at 23:24 UTC and solar wind increased to a maximum of 610 km/s at 23:09 UTC, once more the enhancement was observed at 8 of September at 11:21 UTC to a maximum of 18 nT while the Bz component went southward to a maximum of -17 nT. Geomagnetic sudden impulses of 21 nT (Fredericksburg magnetometer) were observed at 6 of September at 23:48 UTC and 70 nT at the end of the next day with the arrival of both CMEs. In addition, the Earth atmosphere experienced an influence of extraordinary flares (e.g., the M5 flare on 4 of September, X9 flare on 6 of September and the X8 flare on 10 September). The complicated situation before and after the 8 of September, influence of two CMEs gave a rise for some doubts of the referees, if the 8-days running mean is an appropriate measure for the comparison. My suggestion is to discuss in more details important aspects/display and consequences of the event taking into account significant dependences of the ionospheric response at different locations.

Thank you very much for the detailed description as to what happened before and after the storm, I really appreciate it. Indeed, this is a complicated one. As I replied to referee #2, I ran my code using as well a 9-days or 10-days running window and the results are indistinguishable (please, see attached figure 1 and 2). Thus I think there is no need to have doubts on the results. Moreover, very recently, a new study (Sotomayor-Beltran, International Journal of Geophysics, vol. 2018, 2018) has used a 8-days running window showing good results. Hence, I believe in this case as well the statistical method used should not rise any doubts. In the references citing the statistical method ("Data and methods" section), the work from Sotomayor-Beltran 2018 was added as well.

As for the paper Edemskiy et al. Ann. Geophys. vol 36, pp. 71-79, the authors were discussed particularly the anomalous feature which was observed at higher latitudes of the Southern Hemisphere (please, see the area indicated by the black ellipse in the middle panel of the figure below). If you see in your data some similar phenomenon, then it would support the finding published in the Edemkiys paper.

Figure 1: Differential VTEC map for September 7 using 9-day running window

02:00 UT

[Figure]

Figure 2: Differential VTEC map for September 7 using 10-day running window

02:00 UT

[Figure]

Yes, you are correct, Edemskiy et al. Ann. Geophys. vol 36, pp. 71-79, 2018 paid particular importance to the higher latitutde LTE. However, in view of the new recent study (Sotomayor-Beltran, International Journal of Geophysics, vol. 2018, 2018), that supports also the finding of Edemskiy et al. Ann. Geophys. vol 36, pp. 71-79, 2018 and does not present a high latitude LTE, only one at $\sim 44°$S, I believe the findings for the September 8, 2017 storm will undoubtedly provide further support to the work of Edemskiy et al. Ann. Geophys. vol 36, pp. 71-79, 2018. The LTE that appeared during the April 20, 2018 storm was also mentioned in the "Introduction" and "Results and Discussion" sections

Response to Referee #1

Firstly I would like to sincerely thank the referee for his/her valuable comments in an effort to improve my article.

In the revised version I have already addressed all the concerns of referee #1:

1) Total electron content (TEC) enhancements during ionospheric storms of 2017 and 2015 are analyzed in the paper. The author uses his own method of determining deviations in TEC during a storm from quiet conditions. In the majority of ionospheric storm studies, the deviations in foF2 or TEC are studied comparing observed values of the studied parameter with its values during the preceding quiet days, or with a median. The author presents a brief discussion of the method used in the paper (Section 2), however the description is not clear. As far as I understand, for each spatial sell of the data, the 8-day running window is used to calculate the median (X). However, the median is not mentioned later in the text. The formulae (1) and (2) for the upper and lower bounds (UB and LB, respectively) relate UB and LB to $\mu$ and $\sigma$ (UB = $\mu + \sigma$ and LB = $\mu$ $\sigma$), ... were $\mu$ and  are the mean and standard deviation, respectively. One could understand form this determination that $\mu$ is a mean deviation. However then formulae (1) and (2) became senseless, because UB and LB would have a dimension of errors, but not of absolute values of TEC. If the author means that $\mu$ is a mean value, then it is not clear how it has been obtained. Probably, X should stand in formulae (1) and (2) instead of $\mu$. Then at least, the formulae would be understandable.

The brief description of the statistical method indeed is not that understandable as it appears in the paper. But the method (equations) I am following and which I implemented in my software are the ones used and shown in detail in the paper of Zhu et al., 2010. In view of this, I will change lines 19-21 in page 2 with the following text to keep equations (1) and (2) as they appear in the manuscript: "... Li et al.,2015). Assuming that for each cell or line-of-sight the VTEC follows a Gaussian distribution, the mean ($\mu$) of the 8-day VTEC and its associated standard deviation ($\sigma$) are calculated in order to define the upper and lower bounds:". If desired I can add the exact calculation of the mean and the standard deviation. However, this will be basically the same as the ones that appear in the paper of Zhu et al., 2010.

2) The description of the results begins from an error. In the first paragraph of Section 3, Figure 1 is considered. In this paragraph, March 7 and March 8 are mentioned while considering this figure. However, it follows from the caption to Figure 1 that the figure contains data for September, 2017. Obviously, March 7 and March 8 in the text should be September 7 and September 8, respectively.

This is correct. I have already changed in lines 17 and 19 of page 3 the month of March with the month of September (the correct one).

3) Figure 1 does not have dates at the abscissa (only numerals 2), so it is impossible to relate the behavior of geomagnetic and interplanetary indices to UT and dates and to compare this behavior with the TEC data shown in Fig. 2.

This is correct. In the manuscript version ready to download from the AN-GEO website these dates are missing. However, I noticed this a couple of days after my manuscript went online. Reason why, I posted the corrected figures as a comment on July 13, 2018 (which is online in the interactive discussion area). I believe this was a problem with the font types. Now, I am using ones that do not disappear. All figures are now complete.

4) Besides the comments made above, I think that a figure similar to Figure 1 should be included for the March 2015 storm in order to make it possible to compare the data in Fig. 4 with the behavior of geomagnetic and interplanetary indices.

A new figure has been produced (attached to this reply) and its description added to the paper.

5) The language of the paper is poor and needs a serious improvement.

It is correct and my sincere apologies because my mother tongue is not English. I have once again thoroughly checked for typos or gramatic mistakes, and all that needed to be changed has been corrected in the revised version.

Figure 3: The Dst and Kp geomagnetic indices and the southward interplanetary magnetic field (B z ) for the month of March 2015. The vertical red dashed line in all the plots indicates the day that the 2015 St. Patricks day storm occurred (March 17, 2015).

[Figure]

Response to Referee #2

Firstly I would like to sincerely thank the referee for his/her valuable comments in an effort to improve my article.

In the revised version I have already addressed all the concerns of referee #2:

1) Abstract A common response to geomagnetic storms due to the southern vertical interplanetary magnetic fiel (Bz) is the enhancement of the electron density in the ionosphere. This statement general is incorrect. Not all ionospheric storms start with a positive phase. The storm pattern depends on season, longitudinal sector, the intensity of a geomagnetic storm, LT of a magnetic storm commencement.

Yes the statement as it is, is incorrect. On the other hand, I am well aware that the storm pattern depends on season, longitudinal sector and the intensity of a geomagnetic storm. Thus to properly express the statement, I have changed "A common response to ..." to "One of the responses to ..."

2) Looking at the figure 2 I see that the increase of TEC is more extended versus the mid-latitude in the Southern hemisphere but it is possible to see it also in the northern hemisphere. So for my point of view is not a localized enhancement .Looking at the paper by Lei et al. (2017 - 0.1029/2017JA02516) that analyze the same event with TEC and ionosonde data and in their figures it can be seen an increase of differential TEC for all latitudes in the first hours of 8 September, that corresponds to the storm main phase, in the northern hemisphere and in the southern also( but they arrive at 24 degree in latitude)that correspond to a positive ionospheric storm. In the following days a negative ionospheric storm occurred from higher to lower latitude. So it is a typical ionospheric storm with a positive and negative phase , the physical mechanism for the positive phase occurred at 02UT(daytime) that the author identify as a localized event seeing only the figure at 2UT could be due to an expanded convection electric field dur- ing geomagnetic storms in these cases frequently it is observed dTEC enhancements in the mid latitude dayside ionosphere but more investigations are necessary.

As you are pointing out more investigations are necessary for the "expanded convection electric field dur- ing ...". I am also indiating in the very last sentence of the conclusion section that further observations are necessary because the meachnism I am putting forward (a contribution of the super-fountain effect) is not my definite conclusion. Based on Fig.3 from Edemskiy et al. Ann. Geophys. vol 36, pp. 71-79, 2018, if you could observe at the LTE (plume) located at 35°S, this LTE has an extended shape along the mid-latitude, which is similar to the one I am indicating for the September 8, 2017. Hence, following the published results from Edemskiy et al.2018, it is my point of view that the extension south of the EIA during the September 8, 2017 storm is a LTE.

3) L 6 P1 What is the G4 storm? G4 should be explained

I removed G4 from L6 P1 and the title and explaining what it is in the first paragraph of the "Results and dicussion" section.

4) L8 P1 what it is was unexpected Grammar

Thank you very much I corrected this to: what it was unexpected

5) L26 P1 Global Navigation Satellite System (GNSS) receivers, due to its global cover- age, are used as one of the main tools for ionospheric studies. This is not so. The whole morphology of ionospheric storms has been obtained and understood using the world-wide ground-based ionosonde network observations. Only vertical ionospheric sounding gives directly electron concentration in the ionospheric layers. VTEC on one hand is obtained from slant TEC observations on the other hand it includes the plasma- spheric part which is not related to the underlying ionosphere. For this reason VTEC may be only considered as a complementary source of information for such type of analysis.

Yes, I am aware that the ionosondes provide directly electron concentration in the ionospheric layers. However I guided myself from the paper of Hernandez-Pajares et al, J Geod vol 83, pp. 263-275, 2009: "The IGS VTEC maps: a reliable source of ionospheric information since 1998". I have then changed in L26 P1: ".. are used as one of the main tools for ionospheric .." to " .. are used as one of the tools for ionospheric .."

6) L 18 P2 . . .we apply a running window of 8 days. . . Why 8 day window? What is the idea for such choice? What to do with such background if these 8 previous days were disturbed?

What I have seen in several works is 10 day window (Liu et al., Ann Geophys., 2004; Hasbi et al., NHESS 2011; Li et al., Geodesy and Geodynamics 2015; Sharma et al., Quaternary International 2017). I have actually ran my software for 8, 9, 10 ,11, and 12 days and the results of the maps were quite indistinguishable. I have chosen a 8 day window due to the quantity of IONEX files I downloaded from CODE, and which allowed me to see the behavior of the ionosphere in DVETC maps 4 days after the storm. If it is a wish I could changed the maps to the result I get from a 10 day window, but as I mentioned there won't be a noticeable change.

7) L 17 P3 Figure 1 shows that Kp = 8 during the last 3 hours (UT) of March 7 and the first three hours of March 8. March has not been discussed yet in the paper.

Thank you very much for pointing that out. I corrected from "March" to

"Spetember". It was also one of the concerns from referee #1.

8) A positive storm phase (the first phase) of a two-phase ionospheric storm is a normal reaction of the day-time mid-latitude ionosphere to a strong geomagnetic storm (started in the daytime sector). Some examples and mechanisms may be found in J. Atmos. Solar-Terr. Physics., 81-82, 59-75, 2012. Two types of positive disturbances in the daytime mid-latitude F2-layer: Morphology and formation mechanisms.

L 1 P 5 Looking at the figure 2 I see that the increase of TEC is more extended versus the mid-latitude in the Southern hemisphere but it is possible to see it also in the north- ern hemisphere. So for my point of view is not a localized enhancement .Looking at the paper by Lei et al. (2017 - 0.1029/2017JA02516) that analyze the same event with TEC and ionosonde data and in their figures it can be seen an increase of differential TEC for all latitudes in the first hours of 8 September, that corresponds to the storm main phase, in the northern hemisphere and in the southern also( but they arrive at 24 degree in latitude)that correspond to a positive ionospheric storm. In the following days a negative ionospheric storm occurred from higher to lower latitude. So it is a typical ionospheric storm with a positive and negative phase , the physical mechanism for the positive phase occurred at 02UT(daytime) that the author identify as a localized event seeing only the figure at 2UT could be due to an expanded convection electric field dur- ing geomagnetic storms in these cases frequently it is observed dTEC enhancements in the mid latitude dayside ionosphere but more investigations are necessary.

This concern is basically the same as concern 2) in page C2 of you interactive comment. Hence I give the same reply that I give to comment 2):
As you are pointing out more investigations are necessary for the "expanded convection electric field dur- ing ...". I am also indiating in the very last sentence of the conclusion section that further observations are necessary because the meachnism I am putting forward (a contribution of the super-fountain effect) is not my definite conclusion. Based on Fig.3 from Edemskiy et al. Ann. Geophys. vol 36, pp. 71-79, 2018, if you could observe at the LTE (plume) located at 35°S, this LTE has an extended shape along the mid-latitude, which is similar to the one I am indicating for the September 8, 2017. Hence, following the published results from Edemskiy et al.2018, it is my point of view that the extension south of the EIA during the September 8, 2017 storm is a LTE.

In general the paper does not present either any new morphological effect or physical interpretation. I cannot recommend this paper publication.

As in the paper of Edemskiy et al. Ann. Geophys. vol 36, pp. 71-79, 2018, my work presents a new result in the area of LTEs, so I am convinced that it can provide further insights to the community working in this field. I believe as well there is some physical interpretation to my results. One of them, is

that looking into Fig. 3 for the day of the storm (Spetember 9, 2018) the EIA increases, as a consequence of the super-fountain effect, and the crests expand towards the poles. I guess it would have been a breakthrough if for instance the crests would have moved towards the magnetic equator, but still I believe there is a physical interpretation in my work.

[revised manuscript text omitted]

---

## Referee Report (RR1)

The article is dedicated to investigation of localized TEC enhancement during G4 magnetic storm of Sep 8, 2017.

The topic and the obtained results are quite interesting, however there are several remarks to be considered.

First of all, the presented paper is not the first published results of TEC analysis for 8.09.2017. See the paper of J. Lei et al. (DOI: 10.1029/2017JA025166) and a report of D. Horozovic (DOI: 10.13140/RG.2.2.33749.73442).

The title shows that the article investigate TEC during 8.09.2017 but 2 out of 5 figures and almost a half of the Results section text are dedicated to St. Patrick's storm. Either reflect it in the title or reduce St. Patrick's part adding more information about storm from the title.

How did you check the effectiveness of the presented method in LTE detection? It should be shown that it gives the claimed detection of 95%.
According to the text using the method you detected LTE which is turned out to be the southern crest of EIA. Here is CODE GIM map for 18 UT of Mart 17, 2015 with the clear LTE near Weddell sea. Why did not you mentioned its presence? Is it due to absence of a significant ΔTEC variations?

[Figure]

I also would recommend to show a series of ΔTEC maps to present the LTE dynamics more clearly.

Writing in conclusions about "increment in intensity for this LTE" what level do you use as a background? One can think that you mean that LTE exists all the time and became visible increasing its intensity.

Two out of three paragraph of conclusion are dedicated to St. Patrick's storm LTE and the one detected by other authors at August 15, 2015, with presentation of their suggestion of negative Bz influence on LTE generation. It would be better to describe in more details your statement of LTE generation connection with fountain effect and gives some specific details of the investigated LTE manifestation.

Figures 1 and 4 present data for a whole month whereas author uses only several days to analyze. Moreover such a long series makes impossible to see details of indices variations and to check the timestamps presented in the text. Remove the data you are not using and add hours to timescale. Since you compare Dst and Bz dynamics in the text, it would be better to place Kp panel at the bottom of the figure.

Figure 1 label: Bz is a north-south component of IMF and only negative values correspond to southward direction.

Figures 2 and 5. TEC maps are discussed before ΔTEC ones. To make the reading less confusing it would be better to place TEC to the left and ΔTEC to the right. To make ΔTEC maps more contrast it would be better to use some other color map with white in a middle.

Figure 3. Please use the same style labels. Replace square brackets with round ones and add TEC before "TECU". Why do you use fractional values for latitude? It will be easier to read integer numbers.

**pg 3 ln 12:** "5-" instead of "-5"
**pg 3 lns 28-30** This is a well-known fact, but here it is formulated as some new or at least unusual result.
**pg 4** The top paragraph takes 4/3 of the page and is difficult to read. Split it.
**pg 4 lns 10-11** I did not understand why do you duplicate here the information presented above.
**pg 4 ln 23., pg 5 ln 20, pg 6 ln 13:** According to Space Weather storm of August 15, 2015 was rated as G3 (https://www.spaceweatherlive.com/en/archive/2015/08/15). Where did you get G2?
**pg 5 ln 8:** "several" not "some"
**pg 5 ln 10:** replace 4 with 5 in "Fig. 4"
**pg 5 lns 11-12:** What do you mean by "increment of decrement of VTEC"? If it is not about a temporal variations it would be better to describe them as deviations from medium value.
**pg 6 ln 10:** "What is was discovered was..." is not a good formulation. Rewrite this part.

That looks like you are citing the paper of Hathaway and Upton only to explain what do you mean by solar cycle 24-25 minimum. I should note, that mentioned authors write more correctly: "from early 2016 to the end of 2019 – near the expected time of Cycle 24/25 minimum". I recommend to replace "solar cycle 24-25 minimum" by "since 2016".

---

## Referee Report (RR2)

Referee report on the paper "Emergence of a localized total electron content enhancement during the G4 geomagnetic storm of September 8, " byCarlos Sotomayor-Beltran

The paper is devoted to the study of the ionospheric storm, using total electron content data occurred on 7-9 September 2017. In particular the author put in evidence what he calls a localized total electron content enhancements, and increase of TEC respect a background,  at Southern mid latitude hemisphere.

General Comments
The principal comments have not been clarified.

The reply of the author that the same effect has been found in another paper is not an answer.
1) The storms studied are different
2) In the paper of Edemskiy et al.2018 they analysed TEC but also foF2 data .
3) The background that they used is not calculated considering 8 days

At first the author should change the background, secondly he has to analyze ionosonde data.
This spot with increased TEC covers Australia and it is possible to check this increase using Australian ionosonde stations.
02UT was a daytime in the Australian sector and the  NmF2 increase due TAD moving equatorward is a standard situation in the beginning of a strong geomagnetic disturbance.  This should be seen Checking  ionosonde data.

Only after this it is possible to state that that was a localized enhancement.

.So an additional analysis may be recommended (major revision) using Australian ionosonde observations.

---

## Referee Report (RR3)

The author corrected the article in accordance with the referees reviews. The discussion section was substantially improved. But there are several technical corrections to be made:

What was changed in line 21 (pg. 1)? Do you removed all the references?

The statement "By analyzing the latitudinal profiles, it could be determined that the increment of TEC to produce this LTE was of 241.2%" (pg 7, lines 5-6) makes no sense. As far as I understand, LTE is a TEC increment. If you mean to show the LTE intensity, just present TEC value. Currently it is unclear what do you take as a 100%.

Figures 1 and 5 are to be edited since they take a significant space and do not give too much information. Add a grid and more frequent tick marks to Y-axes for Bz and Dst panels.

Pictures of differential VTEC (figs. 2 and 3) looks better, but need higher contrast. Try to use colormap 'seismic' from matplotlib package.

---

## Editor Decision (ED1)

Dear Dr. Sotomayor-Beltran,

Thank you very much for submitting your paper to the Annales Geophysicae, as well as for valuable work you did when preparing the manuscript. It is well known fact, that geomagnetic storms could provide different manifestation and courses in the Earth's upper atmosphere depended on a number of factors and initial conditions. This is a reason, why observations and analysis of the atmospheric response to storm-induced disturbances are very useful and informative for the scientific community.

Now I am coming back to you on the status of your paper. As you already know, decision of the first referee is major revision, the second referee evaluated the manuscript as not acceptable for publication. Both referees are working for the long time in the field and provided very valuable comments, and I believe that their comments will be useful also for your future investigations.

From my side, I would like to point out that the space weather event you analysed, was very interesting, but complicated. Four days-long period before the storm maximum was under the diminishing influence of a positive polarity coronal hole high speed stream (CH HSS), when solar wind speed ranged from 430 to 680 km/s with total field between 3-9 nT. According to the warning issued by NOAA, the geomagnetic field was already at the active levels on 5-6 September. Total field increased twice, for the first time it increased to 16 nT at 6 of September at 23:24 UTC and solar wind increased to a maximum of 610 km/s at 23:09 UTC, once more the enhancement was observed at 8 of September at 11:21 UTC to a maximum of 18 nT while the Bz component went southward to a maximum of -17 nT. Geomagnetic sudden impulses of 21 nT (Fredericksburg magnetometer) were observed at 6 of September at 23:48 UTC and 70 nT at the end of the next day with the arrival of both CMEs. In addition, the Earth atmosphere experienced an influence of extraordinary flares (e.g., the M5 flare on 4 of September, X9 flare on 6 of September and the X8 flare on 10 September). The complicated situation before and after the 8 of September, influence of two CMEs gave a rise for some doubts of the referees, if the 8-days running mean is an appropriate measure for the comparison. My suggestion is to discuss in more details important aspects/display and consequences of the event taking into account significant dependences of the ionospheric response at different locations.

As for the paper Edemskiy et al. Ann. Geophys. vol 36, pp. 71-79, the authors were discussed particularly the anomalous feature which was observed at higher latitudes of the Southern Hemisphere (please, see the area indicated by the black ellipse in the middle panel of the figure below). If you see in your data some similar phenomenon, then it would support the finding published in the Edemkiy's paper.

[Figure]

As the paper contains results, which contribute to our knowledge on the manifestation of magnetic storms, I suggest a major revision. Please, consider very carefully and discuss in the revised version of the manuscript comments of both referees. Both of them (eventually some additional) and me will revise the manuscript again.

Kindest regards

Yours cordially

Dalia Buresova

---

## Author Response (AR2)

Response to Referee #1

Firstly I would like to sincerely thank once again the referee for his/her valuable comments in an effort to improve my article.

In the new revised version I have already addressed all the concerns of referee #1:

The reply of the author that the same effect has been found in another paper is not an answer. 1) The storms studied are different 2) In the paper of Edemskiy et al.2018 they analysed TEC but also foF2 data. 3) The background that they used is not calculated considering 8 days

1) Dear referee, as there are not too many papers explicitly describing LTEs, only Edemskiy et al. Annales Geophysicae, 2018 and Sotomayor-Beltran International Journal of Geophysics, 2018 which study response to a G3 and G2 storm, respectively, another storm of the same category (G4) was looked for. Unfortunately the St. Patrick's day G4 storm did not show any LTEs. 2) Indeed they analyzed GIMs and foF2 data, but they did not analyzed further the GIMs, for example by applying the technique I am using in my paper. Hence, I am sure my work presents potential useful results for the community. 3) As just mentioned Edemskiy et al. Annales Geophysicae, 2018 did not use any background as they did not produce $\Delta$TEC maps. They only used GIMs along with geomagentic indices and other parameters, like f0F2 data, to indicate the existence of their detected LTE; not even a originative mechanism is suggested. In my study I provide a further analysis of the GIMs which is the generation of the $\Delta$TEC maps. On the other hand I have run my software for 10 and 12 days window and the results after a thorough inspection are the same as in Figure 2. It is worth mentioning, I hope, that Zhu et al. 2011 Geodesy and Geodynamics 2, 61-65 and other related studies about ionospheric anomalies have used a time window of 10 days in TEC maps obtaining reliable results.

At first the author should change the background, secondly he has to analyze ionosonde data. This spot with increased TEC covers Australia and it is possible to check this increase using Australian ionosonde stations. 02UT was a daytime in the Australian sector and the NmF2 increase due TAD moving equatorward is a standard situation in the beginning of a strong geomagnetic disturbance. This should be seen Checking ionosonde data.

Dear referee, as I indicated above, a window of 10 or 12 days does not produce any observable difference. In regards to ionosonde data, I have found out that some really interesting ones would be the ones from Canberra, Hobart and Macquarie Is. Unfortunately in the website of the Space Weather Services from the Australian Government Bureau of Meteorology only public data is available until 2014 and also unfortunately I do not have access to other private data. However here in Peru, I am already in talks with a founding member of

the Low-Latitude Ionosphere Sensor Network (Valladares and Chau 2012 Radio Science, 47) which do not only own GPS stations but also Ionosondes and it is very likely that for a follow up study I may use their data. Back to the LTE from September 8, 2017 I have found a work by Lei et al. 2018 where they actually used among a diversity of instruments, ionosondes, to analyze the response to the geomagnetic storm of September 8, 2017 in the Asia-Australian region. After analyzing the ionosondes' data (NmF2 and hmF2) they could see and enhancement in the TEC, as you are also suggesting, and they also attribute this effect not only to TADs but possibly also to PPEFs; hence a combined effect. This last bit has been added to the paper in sections 3.5 and 4, and in the abstract as well.

Response to Referee #2

Firstly I would like to sincerely thank once again the referee for his/her valuable comments in an effort to improve my article.

In the new revised version I have already addressed all the concerns of referee #2:

1) First of all, the presented paper is not the first publihed results of TEC analysis for 8.09.2017. See the paper of J. Lei et al. (DOI: 10.1029/2017JA025166) and the report of D. Horozovic (DOI: 10.13140/RG.2.2.33749.73442).

Dear referee, indeed, this paper is not he first results; hence in the Abstract "first results" was changed to "results". Also the reference of Lei et al 2018 has been added to the text in section 3.1, 3.5 and 4.

2) The title shows that the article investigate TEC during 8.09.2017 but 2 out of 5 figures and almost a half of the Results section text are dedicated to St. Patrick's storm. Either reflect it in the title or reduce St. Patrick's part adding more information about storm from the title

Dear referee, in order not to change the title, the St. Patrick part was reduced (to one subsection) in section 3 and more information about the September 8, 2017 was added (4 subsections) in the same section. Very relevant information about the generation mechanisms was added as well in subsection 3.5

3) How did you check the effectiveness of the presented method in the LTE detection? It should be shown that it gives the claimed dtection of 95%. According to the text using the method you detected LTE which is turned out to be the southern crest of EIA. Here is CODE GIM map for 18UT of March 17, 2015 with the clear LTE near Weddell sea. Why did not you mentioned its presence? Is it due to absence of significant $\Delta$TEC variations?

Dear referee, by using $2\sigma$, it is considered that te confidence level of the detections are at 95%, which has been already indicated and checked by for example Zhou et al 2009 J. Atmos. Sol. Terr. Phys. 71, 959-966; Zhu et al. 2010 Geodesy Geodyn. 1, 23-28; Yao et al 2012 Chi. Sci Bull 57, 500-510. On the other hand, TEC enhancement in the Weddell sea is not mentioned because even though there is a variability, this variability is not outside the $2\sigma$ tolerance under consideration; hence, this especific anomaly does not show up in the produced $\Delta$VTEC maps.

4) I would also recommend to show a series of $\Delta$TEC maps to present the LTE dynamics more clearly

Dear referee, an additional figure (Fig. 3 in the revised version) was added

showing the dynamics more clearly

5) Writing conclusions about "increment in intensity for this LTE" what level do you use as background? One can think that you mean that LTE exists all the time and became visible increasing its intensity

Dear referee, as indicate in section 2 I am using as a background basically the mean and to detect anomalies the tolerance of $2\sigma$. The part of the sentence you are indicating "increment in intensity for this LTE" was reformulated to "increment of TEC to produce this LTE" in the conlcusion section.

6) Two out of three paragraphs of conclusion are dedicated to St. Patric's storm LTE and the one detected by other authors at August 15, 2015, with presentation of their suggestion of negative Bz influence on LTE generation. It would be better to describe in more details your statement of LTE generation connection with fountain effect and gives some specific detals of the investigated LTE manifestation

Dear referee, the two metioned paragraphs were reduced to one where also the storm of August 15, 2015 is mentioned. Also a better explanation of how the LTE from the September 8, 2017 was produced is indicated in section 3.5 following the work from Lei et al. 2018.

7) Figures 1 and 4 present data for a whole month whereas author uses only several days to analyze. Moreover such long series makes impossible to see details of indices variations and to check the timestamps presented in the text. Remove the data you are not using and add hours to timescale. Since you compare Dst and Bz dynamics in the text, it would be better to place Kp panel at the bottom of the figure.

Dear referee, as per suggested, the range of days shown in both figures were reduced. Additional ticks were added to the x- axes of the Dst and Bz plots. The panel of the Kp index was moved to the bottom of both figures

8) Figure 1 label: Bz is a north-south component of IMF and only negative values correspond to southward direction

Dear referee, indeed you are correct. The captions of Figure 1 and also Figure 4 were corrected to properly indicate the vertical component of the IMF.

9) Figures 2 and 5. TEC maps are discussed before $\Delta$TEC ones To make the reaading less confussing it would be better to place TEC to the left ad $\Delta$TEC to the right. To make $\Delta$TEC maps more contrast it would be better to use some other color map with wite in a middle.

Dear referee, the order of the GIMs and $\Delta$TEC maps were changed as well

as the color map in the ∆TEC maps.

10) Figure 3 Please use the same style labels. Replace square brackets with round ones and add TEC before TECU. Why do you use fractional values for latitude? It will be easier to read integer numbers.

Dear referee, the square brackets and y axis title were changed. Some of the x-tick labels are fractional values due to the increment chosen between the x-ticks. To avoid confusion with the numbers they were rotated.

11) pg 3 ln12: "5-" instead of "-5"

Corrected

12) pg3 lns 28-30 This is a well-know fact, but here it is formulated as some new or at least unusual result.

Indeed, it is a well-known fact, thus the sentence was reformulated.

13) pg 4 The top paragraph takes 4/3 of the page and is difficult to read. Split it.

The Result section was split in subsections, and now is more easy to read.

14)pg4 lns 10-11 I did not understand why do you duplicate here the information presented above

Corrected

15) pg 4 ln 23, pg 5 ln 20, pg 6 ln13 Accodring to Space Weather storm of Augus 15, 2015 was rated as G3. Where did you get G2?

Indeed it is rated G3. The correction was done in the lines and pages indicated.

16) pg 5 ln8: "several" not "some"

Corrected

17) pg 5 ln 10: repalce 4 with 5 in Fig.4

Corrected

18)pg 5 lns 11-12 What do you mean by "increment or decrement of "VTEC"? If it is not about a temporal variations it would be better to describe them as

deviations from medium value.

Yes, it is a deviation from the mean value which I am refering to, but in this case for March 17 and 19 at 02:00UT (Fig 5) there are not ionospheric anomalies visible due to the tolerance of $2\sigma$ used. Hence, the sentence was re-formulated in the text to: "there are no anomalous variations of TEC observed"

19) pg6 ln 10: "What is was discovered was .." is not a good formulation, Rewrite this part.

Corrected

20) That looks like you are citing the paper of Hathaway and Upton only to explain what do you mean by solar cycle 24-25 minimum. I should note, that mentioned authors wrie more correctly: "from early 2016 to the end of 2019 - near the expected time of Cycle 24/25 minimum". I recommend to replace "solar cycle 24-25 minimum" by "since 2016".

Dear referee, this was changed in the Abstract, the last sentence of the Results and Discussion, and Conclusions sections.

[revised manuscript text omitted]

---

## Editor Decision (ED2)

Dear Dr. Sotomayor-Beltran,

We appreciate your response to the referee's comments and corrections you made in the manuscript. We have sent the improved manuscript for the second revision, and now I am coming back to you on the status of your paper. We have already received suggestions and comments on the improved version. For your guidance, the comments are appended below. Two reviewers recommended revising the manuscript again. There are still statements and information given in the manuscript, which need substantiation and clarification. One of the referees is will review the manuscript again.

The paper contains original results potentially useful in ionospheric studies and I will recommend it for publishing in the Annales Geophysicae after you consider the suggestions and the manuscript will undergo additional revision. Please, consider and discuss in the revised version of the manuscript comments of both referees and results of the already published works the referees referred to.

If you are prepared to undertake the work required, please submit a list of changes or a rebuttal against each point, which is being raised when you submit the revised manuscript.

Kindest regards

Yours sincerely

D. Buresova

Referee#1

Referee report on the paper "Emergence of a localized total electron content enhancement during the G4 geomagnetic storm of September 8, " byCarlos Sotomayor-Beltran

The paper is devoted to the study of the ionospheric storm, using total electron content data occurred on 7-9 September 2017. In particular, the author put in evidence what he calls "a localized total electron content enhancements", and increase of TEC respect a background, at Southern mid latitude hemisphere.

General Comments The principal comments have not been clarified.

The reply of the author that the same effect has been found in another paper is not an answer. 1) The storms studied are different 2) In the paper of Edemskiy et al.2018 they analysed TEC but also foF2 data. 3) The background that they used is not calculated considering 8 days

At first the author should change the background, secondly he has to analyze ionosonde data. This spot with increased TEC covers Australia and it is possible to check this increase using Australian ionosonde stations. 02UT was a daytime in the Australian sector and the NmF2 increase due TAD moving equatorward is a standard situation in the beginning of a strong geomagnetic disturbance. This should be seen Checking ionosonde data.

Only after this it is possible to state that that was a localized enhancement.

So an additional analysis may be recommended (major revision) using Australian ionosonde observations.

The article is dedicated to investigation of localized TEC enhancement during G4 magnetic storm of Sep 8, 2017.

The topic and the obtained results are quite interesting, however there are several remarks to be considered.

First of all, the presented paper is not the first published results of TEC analysis for 8.09.2017. See the paper of J. Lei et al. (DOI: 10.1029/2017JA025166) and a report of D. Horozovic (DOI: 10.13140/RG.2.2.33749.73442).

The title shows that the article investigate TEC during 8.09.2017 but 2 out of 5 figures and almost a half of the Results section text are dedicated to St. Patrick's storm. Either reflect it in the title or reduce St. Patrick's part adding more information about storm from the title.

How did you check the effectiveness of the presented method in LTE detection? It should be shown that it gives the claimed detection of 95%.
According to the text using the method you detected LTE which is turned out to be the southern crest of EIA. Here is CODE GIM map for 18 UT of Mart 17, 2015 with the clear LTE near Weddell sea. Why did not you mentioned its presence? Is it due to absence of a significant ΔTEC variations?

[Figure]

I also would recommend to show a series of ΔTEC maps to present the LTE dynamics more clearly.

Writing in conclusions about "increment in intensity for this LTE" what level do you use as a background? One can think that you mean that LTE exists all the time and became visible increasing its intensity.

Two out of three paragraph of conclusion are dedicated to St. Patrick's storm LTE and the one detected by other authors at August 15, 2015, with presentation of their suggestion of negative Bz influence on LTE generation. It would be better to describe in more details your statement of LTE generation connection with fountain effect and gives some specific details of the investigated LTE manifestation.

Figures 1 and 4 present data for a whole month whereas author uses only several days to analyze. Moreover such a long series makes impossible to see details of indices variations and to check the timestamps presented in the text. Remove the data you are not using and add hours to timescale. Since you compare Dst and Bz dynamics in the text, it would be better to place Kp panel at the bottom of the figure.

Figure 1 label: Bz is a north-south component of IMF and only negative values correspond to southward direction.

Figures 2 and 5. TEC maps are discussed before ΔTEC ones. To make the reading less confusing it would be better to place TEC to the left and ΔTEC to the right. To make ΔTEC maps more contrast it would be better to use some other color map with white in a middle.

Figure 3. Please use the same style labels. Replace square brackets with round ones and add TEC before "TECU". Why do you use fractional values for latitude? It will be easier to read integer numbers.

**pg 3 ln 12:** "5-" instead of "-5"
**pg 3 lns 28-30** This is a well-known fact, but here it is formulated as some new or at least unusual result.
**pg 4** The top paragraph takes 4/3 of the page and is difficult to read. Split it.
**pg 4 lns 10-11** I did not understand why do you duplicate here the information presented above.
**pg 4 ln 23., pg 5 ln 20, pg 6 ln 13:** According to Space Weather storm of August 15, 2015 was rated as G3 (https://www.spaceweatherlive.com/en/archive/2015/08/15). Where did you get G2?
**pg 5 ln 8:** "several" not "some"
**pg 5 ln 10:** replace 4 with 5 in "Fig. 4"
**pg 5 lns 11-12:** What do you mean by "increment of decrement of VTEC"? If it is not about a temporal variations it would be better to describe them as deviations from medium value.
**pg 6 ln 10:** "What is was discovered was..." is not a good formulation. Rewrite this part.

That looks like you are citing the paper of Hathaway and Upton only to explain what do you mean by solar cycle 24-25 minimum. I should note, that mentioned authors write more correctly: "from early 2016 to the end of 2019 – near the expected time of Cycle 24/25 minimum". I recommend to replace "solar cycle 24-25 minimum" by "since 2016".

---

## Author Response (AR3)

Response to the Topical Editor:

Firstly, thank you once again for your valuable comments in an effort to improve the paper. Answers to your concerns follow.

The author corrected the article in accordance with the referees reviews. The discussion section was substantially improved. But there are several technical corrections to be made:

What was changed in line 21 (pg. 1)? Do you removed all the references?

Dear editor, actually on more reference was added to the ones already written in line 21 of page 1. What happened is that in the mark-up version produced with latexdiff the references are marked in red and with a line looking like if they were removed. However, if you look carefully there is a blue bracket at the very end of line 21 near to the right border of the page (in the mark-up version). In the last version of the manuscript (version 4) uploaded on January 7, 2019 the references are in line 20 and 21.

The statement By analyzing the latitudinal profiles, it could be determined that the increment of TEC to produce this LTE was of 241.2% (pg 7, lines 5-6) makes no sense. As far as I understand, LTE is a TEC increment. If you mean to show the LTE intensity, just present TEC value. Currently it is unclear what do you take as a 100%.

Dear editor, indeed, LTE is basically a TEC increment. Therefore, the last sentence of the first paragraph of the conclussions section was slightly changed to indicate better this fact. The increment is calculated with respect to the average of a stable day (being the average 100%). For this case an average of the curves for September 7 and 9, 2017 was taken.

Figures 1 and 5 are to be edited since they take a significant space and do not give too much information. Add a grid and more frequent tick marks to Y-axes for Bz and Dst panels.

Dear editor, a grid was added in every subplot of the two figures and also tick marks in the Y-axes for Bz and Dst were made more frequent

Pictures of differential VTEC (figs. 2 and 3) looks better, but need higher contrast. Try to use colormap seismic from matplotlib package.

Dear editor, colormap changed to seismic in figures.2, 3, and 6 as per requested.

[revised manuscript text omitted]